# Sugarcane Mosaic Disease: Characteristics, Identification and Control

**DOI:** 10.3390/microorganisms9091984

**Published:** 2021-09-17

**Authors:** Guilong Lu, Zhoutao Wang, Fu Xu, Yong-Bao Pan, Michael P. Grisham, Liping Xu

**Affiliations:** 1Key Laboratory of Sugarcane Biology and Genetic Breeding, Ministry of Agriculture and Rural Affairs, Fujian Agriculture and Forestry University, Fuzhou 350002, China; luguilong666@126.com (G.L.); wzt1417@sina.com (Z.W.); xufu1219@126.com (F.X.); 2USDA-ARS, Sugarcane Research Unit, Houma, LA 70360, USA; yongbao.pan@usda.gov (Y.-B.P.); michael.grisham@usda.gov (M.P.G.)

**Keywords:** sugarcane mosaic disease, characteristics, identification, control strategy, resistance breeding

## Abstract

Mosaic is one of the most important sugarcane diseases, caused by single or compound infection of *Sugarcane mosaic virus* (SCMV), *Sorghum mosaic virus* (SrMV), and/or *Sugarcane streak mosaic virus* (SCSMV). The compound infection of mosaic has become increasingly serious in the last few years. The disease directly affects the photosynthesis and growth of sugarcane, leading to a significant decrease in cane yield and sucrose content, and thus serious economic losses. This review covers four aspects of sugarcane mosaic disease management: first, the current situation of sugarcane mosaic disease and its epidemic characteristics; second, the pathogenicity and genetic diversity of the three viruses; third, the identification methods of mosaic and its pathogen species; and fourth, the prevention and control measures for sugarcane mosaic disease and potential future research focus. The review is expected to provide scientific literature and guidance for the effective prevention and control of mosaic through resistance breeding in sugarcane.

## 1. Introduction

Sugarcane (*Saccharum* spp. hybrids), the most important sugar and energy crop originating in the tropics, is a perennial, high biomass herb ratoon C^4^ crop. It is widely cultivated in more than 100 countries or regions in the tropics and subtropics, with a total area of about 27 million hectares. The world annual output is about 1.95 billion tons of fresh cane, which provides nearly 80% of sugar, 60% of bioethanol, and a total economic value of 75 billion U.S. dollars (FAO, 2019, http://www.fao.org/faostat/zh/#data/, accessed on 25 May 2021). Furthermore, the pressed cane juice can be used to produce diesel, jet fuel, and other high-value products [1,2]. Sugarcane by-products can also be used for direct-fired power generation, field fertilizers, and culture substrate for fruit tree seedlings [3,4].

Mosaic is one of the main viral sugarcane diseases. Systemic infection is caused by the virus after it invades sugarcane. The incubation period is generally about 10 d, but can be up to 20–30 d. The disease may even manifest in the second year of infection [5]. The disease was first described in 1892 by Musschenbroek [6] in Java as “yellow stripe disease”. Subsequently, it was found in Australia [7], Puerto Rico, the United States [8], and India [9]. In 1920, Brandes identified the disease as a transmissible viral disease that could be transmitted by aphis (*Rhopalosiphum maidis* Fitch) [10]. Summers et al. [11] speculated that the disease started in New Guinea and was introduced into Java from infected sugarcane, and then further spread to the Americas and other countries [12]. So far, mosaic has been widely discovered in most sugarcane planting regions around the world [13,14].

Before the 1990s, scientists generally agreed that mosaic was caused by the *Sugarcane mosaic virus* (SCMV). Since then, the *Sorghum mosaic virus* (SrMV) [15] and the *Sugarcane streak mosaic virus* (SCSMV) [16,17] have been independently classified as the new mosaic-causing viral species by the International Committee on Taxonomy of Viruses (ICTV) based on their molecular characteristics. SCMV and SrMV are distributed worldwide [13,18], and SCSMV mainly exists in Asia, including Bangladesh, China [19], India, Indonesia [20], Pakistan, Sri Lanka, Thailand, and Vietnam [21,22]. More recently, the virus has also been reported in Cote d’lvoire in West Africa [23]. In addition, compound infection incidences with different combinations of SCMV, SrMV and SCSMV were frequently reported [24,25,26].

Sugarcane is an asexually propagated crop. If infected stalks are ratooned or used as propagating material, the virus can accumulate in large quantities. Although viruses transfer slowly between plant cells, they move quickly in vascular bundles, along with the flow of plant nutrients [20,27,28]. As a result, the virus can spread to almost every tissue, even the whole stool [29]. In infected sugarcane plants, chlorophyll is destroyed, photosynthesis is weakened, and growth is significantly inhibited [30,31], resulting in shorter internodes, fewer mill-able stems, shorter roots, and a significantly lower sprouting rate and lower yield of cane stems [32,33,34]. Moreover, the disease also reduces juice content, sucrose content, and the crystallization rate [35], which can ultimately reduce sugarcane yield by 10–50% [36], or even 60–80% [12]. The disease has become a pandemic in many countries or regions, including the United States, China, Cuba, Puerto Rico, Argentina, Brazil, and Australia, causing huge economic losses and even bankruptcies to the sugarcane industries [13,14,37].

In recent years, the prevailing sugarcane cultivars, such as ROC22 and Liucheng 05-136, are highly susceptible to the mosaic disease in China [38,39]. In addition, growers’ long-term use of self-produced propagating material, successive cropping, frequent introduction, and improper production management have contributed to the increasing seriousness of mosaic disease in almost all sugarcane growing areas, with the worst infection rate being as high as 100% in some areas [5,14]. The sugarcane region of India faces the same dilemma [40]. In this paper, the pathogenic characteristics, identification methods, and control strategies of sugarcane mosaic disease were reviewed to facilitate the understanding and precise management by providing a reference for green control and resistance breeding in the future.

## 2. Characteristics of Mosaic Disease

### 2.1. Disease Symptoms

The symptoms of mosaic disease caused by SCMV, SrMV and SCSMV are similar, especially in the middle and lower sections of new leaves. In comparison to healthy leaves (Figure 1a), there are many irregular yellow and green inlays, stripes, or mottles alternate with parallel veins on symptomatic leaves, more clearly visible against the sunlight (Figure 1b). Some are mostly normal green with only a few narrow pale-yellow streaks, some show very obvious whole leaf chlorosis, and the seriously infected leaves turn yellow or yellow white, leaving only a few green islets or a small amount of red punctate necrosis (Figure 1c) [12,13], or the tips of new leaves are abnormally twisted (Figure 1d). Some varieties show cryptic or indistinct phenomena at a high temperature, but the symptoms recur as the temperature drops [41].

### 2.2. Hosts

All three viruses infect sugarcane, sorghum (*Sorghum bicolor* L.), and corn (*Zea mays* L.) [12,13,42]. The natural hosts of SCMV include panicum (*Panicum miliaceum* L.), millet (*Setaria italica* L.), green bristlegrass (*Setaria viridis* L.), Johnson grass (*Sorghum halepense* L.), Sudan grass [*Sorghum sudanense* (Piper) Stapf.], and more than 100 species in 40 genera of the *Gramineae* family [15,43,44,45]. Recent reports show that in nature, SCMV can infect St. Augustine grass [*Stenotaphrum secundatum* (Walt.) Kuntze] [46], Columbus grass (*Sorghum almum* Parodi.) [47], pumpkin [*Cucurbita moschata* (Duch. ex Lam.) Duch.] [48], red-veined prayer plant (*Maranta leuconeura* erythroneura) [49], and canna (*Canna indica* L.) [50]. SrMV can infect Miscanthus (*Miscanthus sinensis* cv.) [51] and cause the typical mosaic symptoms. The hosts of SCSMV include panicum (*Panicum miliaceum* L.), buttercup (*Ranunculus japonicus* Thunb.), Sudan grass, Johnson grass, and some other grasses of the *Gramineae* family [52,53,54].

### 2.3. Transmissions

The primary infectious sources of mosaic mainly include infected plants of sugarcane and other *Gramineae* hosts. In nature, transmission of SCMV and SrMV is primarily by several aphid vectors including *Dactynotus ambrosiae* [55], *Hysteroneura setariae* [56], *Longiunguis sacchari* [57,58], *Rhopalosiphum maidis* [10,59], and *Toxoptera graminum* [60,61,62] in a non-persistent manner. Ants also have indirect transmission effect if they interact actively with aphids in diseased sugarcane fields [5,63]. However, insect borne SCSMV has not yet been detected [53,64,65], although *Triticum mosaic virus* (TriMV) and *Wheat streak mosaic virus* (WSMV), which belong to the same genus and share a high sequence similarity with SCSMV, can be transmitted by wheat curl mites (*Aceria tosichella* Keifer) [16,66]. The three viruses are easily transmitted over a short distance by machines, cutting tools, and juice fluid friction, but transmission over long-distance is mainly through infected materials [12,20]. A diagram of the specific transmission pathway is shown in Figure 2.

### 2.4. Epidemiology

The severity of mosaic disease in sugarcane fields is closely associated with sugarcane variety, infected setts, climatic conditions, and intermediate hosts. Among the six *Saccharum* species, *S. officinarum* are highly susceptible, *S. sinense**, S. spontaneum* [67] and *S. barber* [68] are highly resistant or immune, *S. robustum* [67,68] are susceptible to mosaic. This disease was also found on *S. edule* [69], but the resistance to viruses is still uncertain. However, a recent study showed that most of *Saccharum* species have poor resistance to SCSMV; only three out of eight accessions of *S. robustum* among all of the 210 tested clones of *Saccharum* are identified to be resistant [70]. Generally, sugarcane cultivars with more resistant consanguinity also tend to show stronger resistance [65]. Young sugarcane plants are more susceptible than old, mature plants [71]. Drought and less rainfall environments are beneficial to the reproduction and activities of aphids, which promote the spread of mosaic. However, an extremely hot climate is not conducive to disease transmission, leading to slow virus proliferation, less disease symptoms, and less severe disease incidence [5,13]. In general, mosaic often occurs seriously in weedy or intercropping sugarcane fields [35]. High susceptibility of main varieties, relatively high temperatures and less rain, different planting periods in the same region, long-term rotation, and single variety and long-term successive cropping can all lead to a serious occurrence or epidemic of mosaic [72].

## 3. Pathogenicity Characteristics

### 3.1. Taxonomic Status

SCMV, SrMV, and SCSMV belong to the Potyviridae family, of which SCMV and SrMV belong to Potyvirus. Maize dwarf mosaic virus (MDMV), Johnsongrass mosaic virus (JGWV), Zea mosaic virus (ZnMV), Cocksfoot streak virus (CSV), and Pennisetum mosaic virus (PenMV) are grouped together under SCMV subgroup [15,73]. SCSMV belongs to Poacevirus, as do TriMV and Caladenia virus A (CalVA) [17].

### 3.2. Morphology, Size, and Viability

Similar to all members of the Potyviridae family, the three viruses present non-enveloped, flexuous-filamented viral particles [73]. The basic pathogenic characteristics of the sugarcane mosaic virus is shown in Table 1.

### 3.3. Genome Structure

The genome of SCMV, SrMV, and SCSMV is represented by a positive-sense single-stranded RNA (+ssRNA) of about 10 Kb, consisting of untranslated regions (UTR) at both ends and a single open reading frame (ORF) encoding for a large polyprotein. The viral RNA harbour a genome-linked protein (VPg) at the RNA 5′-terminus and a poly (A) tract at the 3′-terminus [75]. The genome structure of the sugarcane mosaic virus is shown in Figure 3. The polyprotein is processed by the virus-encoded proteases P1-pro, HC-Pro and NIa-Pro into 10 mature functional proteins [73,76]. In addition, SCMV and SrMV encode an additional PIPO [77], and SCSMV encodes P3N-PIPO, which are expressed from the P3 ORF through a +2 or +1 frame-coding slippage mechanism, respectively [78,79].

Table 2 describes the main functions of the proteins encoded by viruses of the Potyviridae family. The PIPO and P3N-PIPO mainly affect the movement of the virus between cell-to-cell movement. P3N-PIPO binds to CI to recruit itself into plasmodesmata to promote intercellular movement of the virus [77,80]. Compared with SCMV and SrMV, a highly conserved motif of “Asp-Ala-Gly (DAG)” was absent in the CP sequence of SCSMV, which is necessary for aphid transmission [81,82]. It is worth mentioning that RNA silencing and RNA silencing repressors are mechanisms of defence and counter-defence interactions between host plants and viruses [83]. HC-Pro is the first strong RNA silencing inhibitor discovered [84], which has multiple targets in the RNA silencing pathway to regulate the accumulation of different siRNAs [85]. Moreover, when it fuses with P1, the expression of P1/HC-Pro and the inhibitory activity is enhanced [86]. The P1 protein of *Poacevirus* also has a silencing inhibitory function, which is more obvious than HC-Pro. However, when HC-Pro is present, the inhibitory activity of P1 on RNA silencing is reduced [87].

### 3.4. Genetic Diversity and Taxonomy

During evolution, SCMV, SrMV and SCSMV accumulated a rich pool of genetic diversity. Before 2000, Summers [106,107] and Summers et al. [11] divided SCMV into 10 strains and sub-strains according to the disease symptoms of sugarcane varieties CO281, CP29-291 and CP31-294. Tippett and Abbott [108] divided SCMV into five strains, namely, A, B, D, E and F, according to the mosaic symptoms of CP31-294, but two strains, A and H, according to CP31-588. According to serological cross-reaction, the evolutionary relationship and characteristics of *CP* nucleotide sequence based on the 3’-end sequence of the viral genome, *Potyvirus* isolates from Australia and the U.S. were divided into 11 SCMV strains, namely, MBD, A, B, D, E, SC, BC, Sabi, ISIS, Bris, and Bund, and three SrMV strains, namely H, I and M [15,43]. Since 2000, the genetic diversity of mosaic viruses isolates from China [109,110], South Africa [111], India [112,113], Mexico [114], Argentina [115] and Thailand [116,117], and others have also been reported based on the nucleotide and/or amino acid sequence variability of *CP* [81,118], *HC-Pro* [119] and *P1* [79]. To sum up the results of genetic diversity, 216 relatively complete nucleotide sequences of the coat protein genes from102 SCMV isolates from 26 countries, 58 SrMV isolates from five countries, and 56 SCSMV isolates from 11 countries are downloaded from the NCBI database (accessed date 16 May 2021) and analysed using the maximum likelihood method (ML) of MEGA v6.0 software (Raynham MA, USA). The results are depicted in Figure 4.

Plant RNA viruses mutate in a variety of ways, including natural selection and substitution, transversion, deletion, insertion, recombination, reassortment, etc. [120,121]. Gell et al. [122] found that recombination was the main driving force for evolving SCMV subgroup populations. Recombination and strong selection pressure may accelerate the elimination process of deleterious mutations in the SCSMV genome *P1* gene [79]. Strong purifying selection has been dominant in Indian SCMV populations, in which the CI and HC-Pro genes are prevalent [123]. He [64] reported that negative selection and genetic drift rather than recombination were the main driving force for the evolution of SrMV and SCSMV in China. In addition, natural selection, gene migration and geographical isolation may also affect the evolution of virus population in different regions [22,81,124].

## 4. Diagnosis/Identification

### 4.1. Visual Observation

Mosaic disease can be identified by visual inspection of sugarcane leaves for symptoms when evaluating germplasm resistance [38,39,125,126]. The method is simple and timesaving. However, it cannot determine the virus types. In addition, it requires a higher level of professional knowledge, skill, and ability to catch sugarcane leaf mosaic symptoms with the naked eye.

### 4.2. Biological Identification

Traditionally, the type of virus, host range and transmission mode are determined mainly based on the symptoms of the host plant after inoculation with the virus [11,127]. Before 1940, Summers EM artificially inoculated leaves of indicator plants of varieties CP31-294 and Co281 with press juice friction from a symptomatic plant and made a judgment based on whether the leaves showed any mosaic symptoms [11,106,107]. However, it is hard to do, labour intensive, and difficult to determine the specific virus strain. The amount of needed plant material can fill up the greenhouses and strict test conditions are required.

### 4.3. Microscopic Observation

The most prominent pathological feature of plant tissue cells infected by a virus of the Potyviridae family is the production of columnar inclusions. The feature has been used to distinguish among virus species. Edwardson [128] divided cylindrical inclusions into four types: scrolls body and pinwheels (Type Ⅰ); lamellar aggregates with arm extension and pinwheels (Type Ⅱ); scrolls body, pinwheels, and lamellar aggregates (Type Ⅲ); and scrolls body, pinwheels, and short and curved lamellar aggregates (Type Ⅳ). Studies have shown that the characteristics of columnar inclusions in SCMV-, SrMV- [129] and SCSMV-infected [53] sugarcane leaf cells were similar to Ⅲ, Ⅰ, and Ⅱ, respectively. However, this method is seldomly used in practice due to the complicated operation procedure of electron microscopy.

### 4.4. Serological Detection

Serological detection is a simple, rapid, and low-cost method for plant viruses, including agar gel immunodiffusion (AGID), electroblot immunoassay (EBIA), double antibody sandwich enzyme-linked immunosorbent assay (DAS-ELISA), direct antigen coating ELISA (DAC-ELISA), indirect ELISA, and dot enzyme-linked immunosorbent assay (Dot-ELISA) (also dot immunobinding assay, DIBA; or dot-bolt immunobinding assays, DBIA) [74,130,131]. Hema et al. [74] detected SCSMV by AGID, DAC-ELISA and EBIA. Mohammadi et al. [132] used DAS-ELISA and DIBA to detect SCMV and SrMV in infected samples. SrMV was first detected in *Miscanthus* by Grisham et al. [51] using indirect ELISA. Gaur et al. [133] used DAC-ELISA and DIBA to detect SCMV in infected cane juice samples even when the samples were diluted to 1/150. Wang et al. [134] established a high-throughput DAC-ELISA method for the detection of SrMV. However, serological detection requires specific antiserums, and its sensitivity is relatively lower than molecular detection techniques [130,131].

### 4.5. Molecular Detection

PCR-based molecular techniques have been developed and widely used in the detection of the mosaic virus, such as direct-binding polymerase chain reaction (DB-PCR), reverse transcription PCR (RT-PCR), immunocapture RT-PCR (IC-RT-PCR), duplex immunocapture RT-PCR (D-IC-RT-PCR), one-step multiplex RT-PCR, real-time quantitative RT-PCR (qRT-PCR) and loop-mediated isothermal amplification (LAMP) [83]. These molecular detection techniques have a high sensitivity and specificity and can identify virus species rapidly and accurately. Yang and Mirkov [135] used DNA restriction enzymes to digest RT-PCR fragments to produce SCMV- or SrMV-specific RFLP patterns without the need of using identification host plants. Xie et al. [136] established a one-step quadruplex RT-PCR method to simultaneously detect four viruses, namely, SrMV, SCMV, SCSMV and *Sugarcane yellow leaf virus* (SCYLV). Smith and Velde [137] were able to detect SCMV by RT-PCR in samples that were diluted 10,000-fold. Fu et al. [52] developed a qRT-PCR method for the detection of SCSMV, which was 100 times more sensitive than conventional RT-PCR. Hema et al. [138] reported that IC-RT-PCR was 5,000 times more sensitive than DBIA and 10 times more sensitive than DB-PCR to detect SCSMV. Chen et al. [139] demonstrated that the sensitivity of detecting SrMV by IC-RT-PCR 100-times higher than that of indirect ELISA and 1000-times higher than that of Dot-ELISA and therefore recommended IC-RT-PCR was more suitable for large volumes of samples. Subba et al. [140] developed a D-IC-RT-PCR method that combined both serological and molecular methods to simultaneously detect and distinguish SCMV and SCSMV and was more sensitive than DAC-ELISA. In addition, a new reverse transcription-LAMP (RT-LAMP) technology was also applied to detect SCMV and SrMV with a lower sensitivity than RT-PCR and qRT-PCR [141]. The primer sequence information of different molecular techniques for detecting sugarcane mosaic viruses is shown in Table 3.

## 5. Prevention and Control Strategy

### 5.1. Exploitation and Utilization of Resistant Germplasm

Selection and rational distribution of disease-resistant varieties are the most economic and effective prevention and control measures against these viruses. The genetic base of modern sugarcane cultivars is narrow, with about 80% from *S. officinarum*, 10–15% from *S. spontaneum*, and 5–10% from recombinant chromosomes [142]. During the long evolutionary process, the *Saccharum* and its related genera have formed extremely rich and valuable germplasm resources [68,143], which contain large numbers of disease resistance genes [144]. In recent years, some *Saccharum* hybrid clones with high resistance to mosaic disease have been identified by natural infection or artificial inoculation, such as SP70-1143, IACSP95-2078 [145], SWSwm1 [146], YG34, YG55 [147], ROC16 [38], GT03-2309, and LC03-1137 [39]. The current artificial inoculation methods are shown in Table 4.

### 5.2. Acceleration of Molecular Breeding

Molecular marker-assisted breeding and genetic engineering improvements have helped promote the development of resistant varieties. Molecular marker-assisted breeding is an effective method to accelerate the breeding process of multi-resistant varieties. However, the technology is limited by the lack of markers closely linked to disease resistance. Sugarcane (2 *n* = 12 *x* = 100−130; genome size = ~10 Gb) is a highly complex autopolyploid and aneuploid crop, and a complete reference genome of modern sugarcane cultivated species is still lacking up to now [154,155]. The development of molecular markers associated with economic target traits is an extremely slow process for sugarcane. Previous studies on the development of molecular markers for mosaic resistance were mainly focus on corn (2 *n* = 2 *x* = 20; genome size = ~2300 Mb) [156,157,158,159,160,161,162,163,164]. However, these studies may provide a good reference for developing molecular markers and related gene mining on sugarcane. In addition, genetic engineering is an effective way to obtain disease-resistant sugarcane varieties [165]. The most mature strategy is *CP* gene-mediated transfer since the first report of the introduction of the *Tobacco mosaic virus* (TMV) *CP* gene into tobacco in 1986 [166]. Smith et al. [167] used a gene gun to bombard a *SCMV-CP* gene into sugarcane meristem and obtained chimeric transformed plants. Subsequently, Joyce et al. [168], Ingelbrecht et al. [169], and Sooknandan et al. [170] also succeeded in obtaining transgenic sugarcane plants by using the same method. On the other hand, an RNA interference technology that targets sugarcane virus encoded RNA silencing inhibitors has also been successfully used to develop highly effective disease-resistant transgenic sugarcane plants [171,172,173,174,175]. Field experiments on transgenic sugarcane have shown improved resistance to mosaic disease with significantly increased yield and sucrose content [171,176,177,178]. However, none of the transgenic sugarcane has yet been applied in field production due to regulations.

### 5.3. Application of Virus-Free Plantlets

The application of virus-free plantlets through tissue culture can not only eliminate or slow down the incidence of mosaic disease, but also increase the sucrose content by more than 0.5% and the cane yield by 20–40% [179]. In one case, the yield increased by more than 100% [180]. The common methods of detoxification include heat treatment, stem buds tissue culture detoxification, and ultra-low temperature therapy. SCMV elimination can be achieved by cultivation at 52 °C, 57.3 °C, 57.3 °C, once every other day, 20 min each time, without harming cane buds [179]. SCMV can also be removed by cryo-treating micro shoot tips (~3 mm) [181] or by inserting 4~8 mm axillary bud explants into MS regeneration medium supplemented with a 25 mg/L of ribavirin [182]. Hot water treatment for 10 min at 55 ℃ was not effective for SCSMV elimination [183]. Detoxification by stem buds tissue culture could completely eliminate SCSMV [184,185]. A combination of heat treatment and axillary shoot tip culture had a better effect on virus removal [186].

To determine the technical specifications for the production and testing of sugarcane virus-free plantlets, leaves were collected from the virus-free plantlets produced by the heat treatment and axillary shoot tip culture technology in 2015, followed by tracking and PCR testing the leaves of 120 putative virus-free plantlets collected from different companies at different ecological demonstration sites in China [187]. The PCR results showed that SCMV was not detected in 100% of the samples, the rate of SrMV detection was greater than 35%, and the rate of SCSMV detection was more than 50%. Therefore, it was concluded that SrMV and SCSMV were more difficult to be eliminated from infected sugarcane plants than SCMV. Although the virus-free effect of sugarcane stalk can be improved by appropriately increasing treatment temperature and time (30 min at 59 °C), the germination rate, however, was significantly reduced to only about 20% and the water temperature was difficult to control accurately in an industrial setting (unpublished).

### 5.4. Strengthen Cultivation and Control

Good field management is another effective way to enhance plant resistance and reduce the spread of viruses. Specific measures include: (1) avoiding planting virus host crops around or in the sugarcane fields [188]; (2) the timely removal of infected plants and weeds; (3) improvement in soil structure, rational fertilization, and irrigation to promote plant growth and improve disease resistance [189,190]; (4) chemical and biological control of aphids [63]; and (5) fortifying cropping systems and paying attention to rotation with non-host crops, such as soybeans, sweet potatoes and peanuts [191].

## 6. Perspective

The cultivation and planting of resistant varieties are the most economic and effective methods to control sugarcane infecting viruses, and growers are most likely to adopt this technology. However, due to the diversity of pathogens, the highly complex genome of sugarcane, the wide segregation of traits among hybrid progenies, and the extremely low probability of excellent gene aggregation, sugarcane cross-breeding may have to rely on a huge population, for example, 1 to 1.2 million of plantlets to be planted annually in China. Therefore, it is very difficult to select good varieties that have both commercial value and mosaic resistance. In addition, field evaluation tests of virus-free plantlets from integrated detoxification technology in China have shown that the “virus-free” effect is not ideal, even if the plantlets were produced by a more effective method of combining heat treatment and axillary bud or shoot tip culture. In view of all these facts, the development of molecular markers associated with disease resistance genes should be the most effective way to breed sugarcane varieties resistant to different viruses. Sugarcane propagates asexually and F_1_ populations can be used for correlation analysis between genotypes and phenotypic traits. There are also successful cases on genetic engineering via gene gun bombardment or gene editing to improve disease resistance in sugarcane. Successfully edited alleles can be up to 49 copies simultaneously targeting the chlorophyll content gene, which draws a new blueprint for transforming the disease resistance pathway in sugarcane. We hope that the present review can provide scientific references and thoughts for the effective prevention and control of viruses in sugarcane.

## Figures and Tables

**Figure 1 microorganisms-09-01984-f001:**
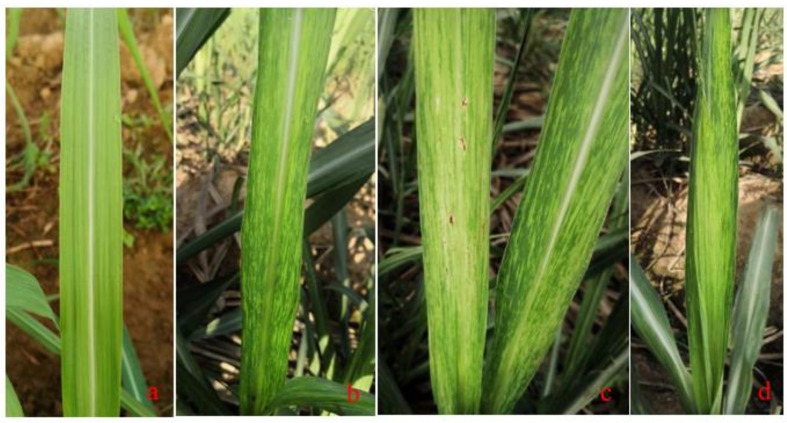
Symptoms of mosaic infection in sugarcane leaves. (**a**). healthy leaves; (**b**). infected leaves; (**c**). severely infected leaves; (**d**). abnormally twisted top leaves of infected plant.

**Figure 2 microorganisms-09-01984-f002:**
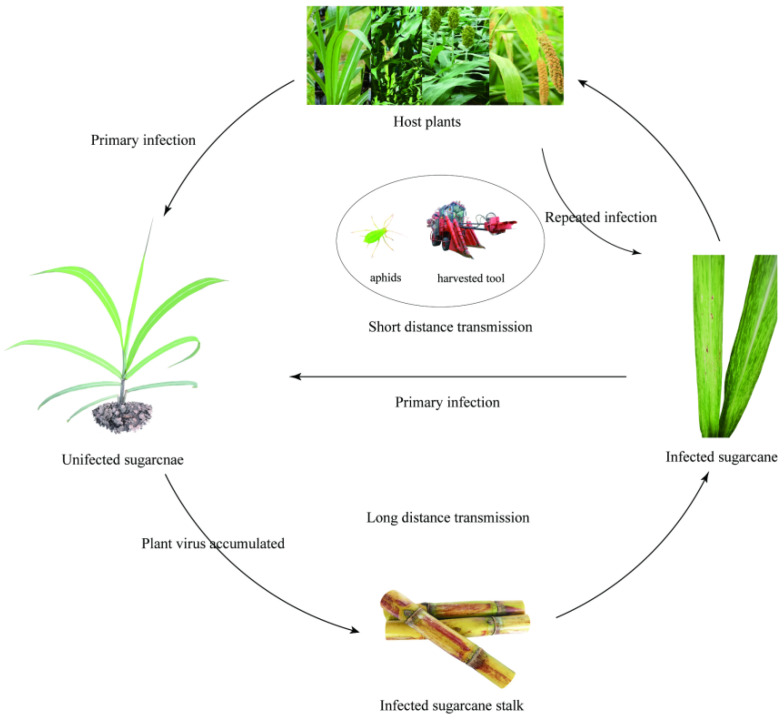
The transmission pathway of sugarcane mosaic disease.

**Figure 3 microorganisms-09-01984-f003:**
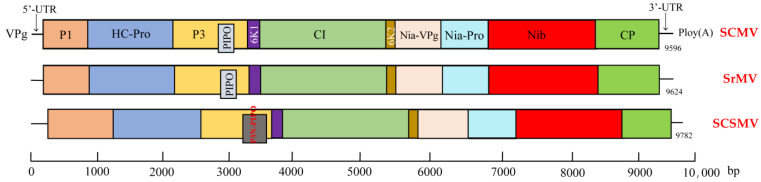
The genome structures of sugarcane mosaic viruses. Note: the reference genomes SCMV, SrMV, and SCSMV were assembled based on NC003398, NC004035 and GQ388116, respectively. P1, Protein 1; HC-Pro, Helper component proteinase; P3, Protein 3; PIPO, Pretty interesting Potyviridae ORF; 6K1, Protein 6K1; CI, Cylindrical inclusion protein; 6K2, Protein 6K2; VPg, Viral protein genome-linked; NIa-Pro, Nuclear inclusion a protein; NIb-Pro, Nuclear inclusion b protein; CP, Coat protein; P3N-PIPO, P3 N-terminus Pretty interesting Potyviridae ORF.

**Figure 4 microorganisms-09-01984-f004:**
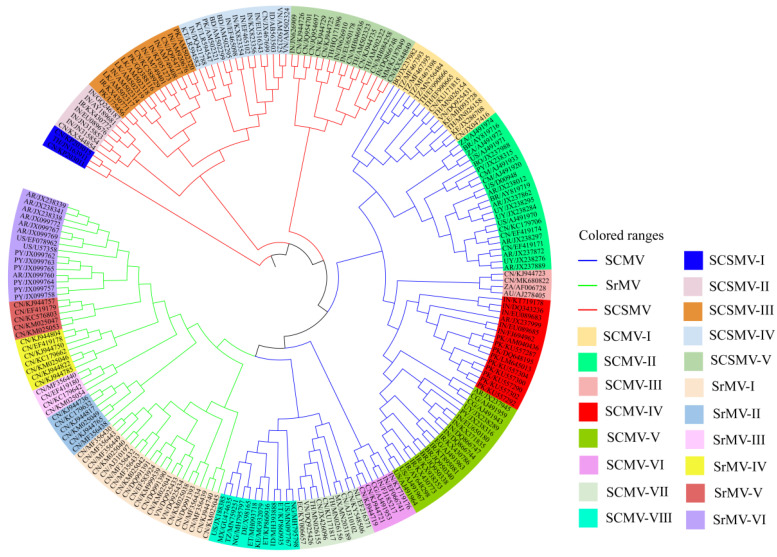
Genetic diversity analysis of coat protein genes of three viruses causing sugarcane mosaic disease. Note: Red line represents SCSMV, blue line represents SCMV, and green line represents SrMV. Different colour boxes represent different subgroups.

**Table 1 microorganisms-09-01984-t001:** Basic pathogenic characteristics of sugarcane mosaic virus.

Virus Species	Virion Size	Inactivation Temperature	Survival Time	Dilution Limit	Standard Sedimentation Constant and Buoyancy Density	References
SCMV	630–770 nm × 13–15 nm	53–57 ℃	in vitro survival time is 17–24 h at 27 °C and 27 d at −6 °C	10^−3^−10^−5^	160–175 S, 1.285–1.342 g/mL	[45]
SrMV	620 nm × 15 nm	53–55 ℃	in vitro survival time is 1–2 d at 20 °C	10^−2^–10^−3^	-	[13]
SCSMV	890 nm × 15 nm	55–60 ℃	in vitro survival time is 1–2 d and 8–9 d at room temperature and 4 ℃	10^−4^–10^−5^	-	[53,74]

Note: “-” means undetermined.

**Table 2 microorganisms-09-01984-t002:** The main role of 12 functional proteins of Potato virus Y.

Protein Name	Role	References
P1	the sequence is the least conserved and its size variability is the highest, participate in virus replication related to the widespread adaptation of the virus to host species	[88] [87]
HC-Pro	RNA silencing repressor, related to viral vector, cellular and long-distance movement participates in virus replication and symptom presentation the expression activity was enhanced by fusion with P1	[89] [90] [86]
P3	participates in replication, accumulation, and cell-to-cell movement of the virus determines host range and symptoms	[91] [92]
PIPO	conserved protein, affects the movement of the virus between cells	[77]
P3N-PIPO	binds to CI, promote intercellular movement of the virus	[80]
6K1	be involved in intercellular movement of the virus participate in virus replication	[93] [94]
CI	it has helicase activity, coping with and overcoming plant defense responses	[95]
6K2	participate in viral replication and intercellular movement be associated with long-distance transportation, symptom induction inducing vesicle formation	[96] [97] [98]
VPg	may act as primers in viral replication affect the cellular movement and long-distance transmission	[99] [100]
NIa-Pro	cleave polypeptide proteins viral replication may be associated with symptoms and host specificity	[101] [102] [103]
NIb-Pro	involved in replication catalyzes the formation of template and prime-dependent poly (U)	[104] [105]
CP	play roles in viral cell-to-cell and long-distance movement, replication, and vector transmission	[73]

**Table 3 microorganisms-09-01984-t003:** The primer sequence information of different molecular detection techniques for sugarcane mosaic viruses.

Technology Name	Detection Virus	Primer Sequence (5′→3′)	Sequence Position	Amplification Size (bp)	Annealing/ Incubated Temperature (°C)	Reference
RT-PCR	SCMV	F: TTTYCACCAAGCTGGAA R: AGCTGTGTGTCTCTCTGTATTCTC	*NIb-CP*	873(-A), 885(-B/-D), 897(-E)	60	[135]
	SrMV	F: AAGCAACAGCACAAGCAC R: TGACTCTCACCGACATTCC	*NIb-CP*	871(-SCH/-SCI/-SCM)	60	
One-step RT-PCR	SCMV	F: CAATCTTGAGGAATGCGGAAAAC R: ATCGATAGGCCCACAAATGAGTCT	*HC-pro*	720	54	[136]
	SrMV	F: ACAGCAGAWGCAACRGCACAAGC R: CTCWCCGACATTCCCATCCAAGCC	*CP*	860		
	SCSMV	F: ATTTTGCCGTCCCGTTTTACATC R: AGCGCGTTGTCTTTCTTCTTCAGTCA	*NIa-NIb*	1160		
qRT-PCR	SCSMV	F: FAM-TGCTGCATTGATTTCGTGATGGTG-TAMRA R: FAM-TGCTGCATTGATTTTGTGATGGTG-TAMRA	*CP*	115	60	[52]
IC-RT-PCR	SCSMV	F: GGACAAGGAACGCAGCCACCTCAG R: TTTTTTCCTCCTCACGGGGCAGGTTGATTG	*CP*	1047	55	[138]
	SrMV	F: ATCGCCATGGCTGCAGGGGTTGGAACGGTGG R: ATCGCTCGAGGTGGTGCTGTTGCACCCCAAG	*CP*	~1000	54	[139]
D-IC-RT-PCR	SCMV	F: ATGTC(GA)AAGAA(GA)ATGCGCTTGC R: -d(T)_18_(AGC)-	*CP*	~900	56	[140]
	SCSMV	F: AAGTGGTTAAACGCCTGTGG R: -d(T)_18_(AGC)-	*NIb-CP*	~1400	56	
RT-LAMP	SCMV	F3-4: GTGGTCTAATGGTATGGTGTATT B3-4: TCTAGCTGGTGTCCTTGAA FIP-4: CCGGAATGTTGGAGATGCGTGTTGGACAATGATGGATGGA BIP-4: TTCAGTGATGCAGCTGAAGCACGCTGAAGTCCATATCGTG	*CP*	-	63	[141]
	SrMV	F3-4: ACAACAACAAGACATTTCAAACA B3-1: GTTCCGATACTCTATGTACGC FIP-4: CATTAATATTAGGTGAGCATCCGTTCTCTAGATGATACGCAGATGACAG BIP-4: TTCAGTGATGCAGCTGAAGCACGCTGAAGTCCATATCGTG	*CP*	-	63	

Note: M = A/C, Y = C/T, W = A/T, R = A/G, and W = A/T in primer sequences. FAM: 6-carboxyfluorescein, TAMRA: 6-carboxy tetramethyl rhodamine. -A, -B, -D and -E indicate different SCMV strains; -SCH, -SCI and -SCM indicate different SrMV strains. “-” means uncertained.

**Table 4 microorganisms-09-01984-t004:** Comparison of artificial inoculation methods for mosaic resistance evaluation.

Type	Methods	Characteristics	Application	References
Airbrush inoculation	The venom was uniformly sprayed on the leaves of sugarcane under high pressure	The operation is simple; but the inoculation efficiency is not high	Larger group material	[148,149,150]
Mechanical inoculation	Use fingers dipped in a little quartz sand containing the disease venom, through the young leaf scratch infection	Strict inoculation conditions; but the work efficiency is not high, the wound is not uniform, the effect of vaccination is not stable	Small group material	[145,146,151]
Pricking or inject inoculation	Use a micro syringe to absorb a small amount of venom and inject it into the axillary buds and subcutaneous tissue of sugarcane species	Venom dosage controllable, standardized use, less damage to plants, the effect of inoculation was higher; but inoculation efficiency mediocre	Moderate group material	[151,152]
Stalk cutting inoculation	Cut off the above ground part of the cane plant with a sharp blade or branch shears. Immediately drop quantitative of virus liquid into the wound and shade it for 24 h	Simple and efficient operation method, the effect of inoculation was higher; but dark treatment environment is difficult in the field, serious damage to plants	Large group material	[147,153]

## Data Availability

All the data is present in the manuscript.

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
