# Peer review of "Sugarcane Mosaic Disease: Characteristics, Identification and Control"

_microorganisms, 2021, doi:10.3390/microorganisms9091984_

Round 1

Reviewer 1 Report

The paper is well written a good summarization of the knowledge of sugarcane viruses and protection against the viruses, containing the detection as well. There are some few mistakes and scientificly not good wording or drafting.

For example there is no such that: "mosaic viruses" only Independent virus species causing mosaic symptoms

R54 "stalk … used as seed" never use for vegetative propagation as seed, becaus it is deceptive, please use propagating material. Specially in virology because there are seed transmission of viruses.

R69 same as R54

R210 "mosaic virus population" use virus population

R223 "a juice liquid friction" use press juice

R280 "for mosaic Disease" use against this viruses

R319-320 "warm water detoxification" detoxification is not the appropriate word use heat treatment, heat therapy or thermo therapy

R321 "SCMV detoxification" use SCMV elimination

R325 same as R321

R343 "spread of disease" Disease can't spread only pathogen can spread use spread of viruses

R343 "mosaic virus hosts"  use virus hosts

R351 "control mosaic Disease" use control of sugarcane infecting viruses, "and sugarcane growers" use growers

R362 "mosaic viruses" use viruses

R369 "mosaic Disease" use viruses

Table3 "mechanical friction inoculation" use mechanical inoculation

Reviewer 2 Report

The manuscript describing characteristics, identification and control of sugarcane mosaic diseases is clear, well written and easy to read and contain mainly accurate information on sugarcane viruses causing mosaic disease. However, I have three remarks, two concerning epidemiology and one concerning table 2:

1/In the epidemiology section, when describing saccharum species resistance, this did not concern SCSMV, rescents results from India showed that 90% of the spontaneum tested are infected by SCSMV (https://doi.org/10.1007/s12355-021-00995-3).

2/ reference 69 sited mosaic in solely the one variety of S. edule cultivated in Papua New Guinea.

3/ Table 2: Protein name and functions are not well alligned, please drop a line between each protein for clarification.

In addition main Chinese’s references are not accessible or not in English.

Reviewer 3 Report

The manuscript is an extensive revision of the mosaic disease of sugarcane and the three viruses associated to this symptomatology and provides an impressive revision of the scientific literature related to this topic.

I have some minor points that from my point of view should be corrected in the manuscript

1-In lines 44-46 the viruses are described “Before the 1990s, the scientists generally agreed that mosaic was caused by Sugarcane mosaic virus (SCMV). Since then, Sorghum mosaic virus (SrMV) [15] and Sugarcane streak mosaic virus (SCSMV)”

After this first introduction, viruses should be referred trough the text as SCMV, SrMV and SCSMV

2-  Line 142

“All three virions are curved linear and unenveloped, similar to members of the Potato  viridae Y (PVY) [72]. “

Suggestion:

Similar to all members of the Potyviridae family the three viruses present non-enveloped, flexuous-filamented viral particles

3- Table 1

Virus species instead of type  

virion size instead of size

inactivation temperature instead of passivation temperature

Dilution limit of what? Minimal concentration of viral particles in the inoculum used for mechanical inoculation?

4-Line 148-155

Suggestion:

The genome of SCMV, SrMV, and SCSMV is represented by a positive-sense single-stranded RNA (+ssRNA) of about 10 Kb consisting of an untranslated region (UTR) at both ends and a single open reading frame (ORF) encoding for a large polyprotein. The viral RNA harbor a genome-linked protein (VPg) at the RNA 5’-terminus and a poly (A) tract at the 3′-terminus. The genome structure of sugarcane mosaic virus is shown in Figure 3. The polyprotein is processed by the virus-encoded proteases P1-pro, HC-Pro and NIa-Pro into 10 mature functional proteins [72, 75]. In addition, SCMV and SrMV encode an additional PIPO [76], and SCSMV encodes P3N-PIPO which are expressed from the P3 ORF through a +2  or +1 frame-coding slippage mechanism, respectively  [77, 78].

5-Legend Figure 3

Suggestion:

Figure 3. The genome structures of sugarcane mosaic viruses. Note: The reference genomes SCMV, SrMV, and SCSMV were assembled based on NC003398, 158 NC004035 and GQ388116, respectively. P1, protein 1; HC-Pro, Helper component proteinase; P3, Protein 3; PIPO, Pretty interesting Potyviridae ORF; 6K1, Protein 6K; CI, Cylindrical inclusion protein; 6K2, protein 6K2; VPg, Viral protein genome-linked; NIa-Pro, Nuclear inclusion a protein; NIb-Pro, Nuclear inclusion b protein; CP, Coat protein; P3N-PIPO, P3 N-terminus Pretty interesting Potyviridae ORF.

6-Line 165

“At present, the functions of the 12 proteins encoded by the PVY are basically known 165 (Table 2)”

Suggestion

In Table 2 are described the main functions of the proteins encoded by viruses of the  Potyviridae family

7-Line 166

“The PIPO and P3N-PIPO mainly affect the movement of the virus between host cells”.

This statement is also indicated in table 2. From my point of view, it would be more accurate to say that The PIPO and P3N-PIPO are implicated the cell-to-cell movement (or intercellular movement or local movement).

8-Table 3 and in the main text

“P1 RNA silencing repressor, related to viral vector and movement “

“HC-Pro participates in replication, accumulation, and movement of the virus”

“PIPO conserved protein, affects the movement of the virus between cells “

It should be specified in which type movement are implicated each viral protein. For example, Hc-Pro is a multifunctional protein implicated in the aphid transmission, replication, and virus cell-to-cell and systemic movement and function as suppressor of RNA-mediated gene silencing. P3N-PIPO is required for aphid transmission, for an efficient cell to cell transport and acts as a suppressor of RNA-mediated gene silencing

9-Line 169

Why “However”?

10-Lines 174-175

“Moreover, when it fuses with P1, the expression of P1/HC-Pro and the inhibitory activity enhanced.

For me this sentence makes no sense

11- “Section 4.5. Molecular detection”

It may be most helpful for readers interested in the topic to include a table with the primer sequences of the primers used to detect the three viruses in the one-step quadruplex RT-PCR simultaneous detection method and or to amplify individual viruses by one of PCR-based molecular techniques described in the text
